# Impact of Ferulated Arabinoxylans from Maize Bran on Farinograph and Pasting Properties of Wheat Flour Blends

**DOI:** 10.3390/foods13213414

**Published:** 2024-10-26

**Authors:** Muzzamal Hussain, Senay Simsek

**Affiliations:** 1Department of Food Science, Government College University, Faisalabad 38000, Punjab, Pakistan; muzzamalhussain24@gcuf.edu.pk; 2Whistler Center for Carbohydrate Research, Department of Food Science, Purdue University, West Lafayette, IN 47907, USA

**Keywords:** dough development, extensibility, ferulated arabinoxylans, maize bran, paste viscosity, water absorption, wheat flour blends

## Abstract

This research explores the extraction of ferulated arabinoxylans (FAXs) from maize bran and their incorporation into wheat flour to assess their impact on rheological and pasting properties. Flour blends were prepared with FAX concentrations of 0%, 2%, 4%, 6%, 8%, and 10%, and these blends were then evaluated using farinograph, mixograph, micro-extensibility, and viscosity analyses. The results indicated that farinograph water absorption increased significantly (*p* ≤ 0.05), ranging from 54.9% to 60.5%, as the FAX content rose, correlating with higher gel-forming potential. Notably, the 2% FAX treatment (FAX2) exhibited the longest dough development time at 24.7 min. Stability and mixing time index (MTI) values also varied significantly (*p* ≤ 0.05) among treatments, with FAX2 displaying a longer mixing time of 14.4 ± 3.3 min. A pasting analysis revealed a significant decrease in peak and hot paste viscosity (*p* < 0.05) with increasing FAX concentrations, suggesting an association between lower hot paste viscosity and reduced breakdown. Micro-extensibility measures further indicated that blends with 0% and 2% FAX had greater extensibility, while the 4% FAX blend showed higher resistance. Overall, this research aims to advance the understanding of how these components can enhance flour functionality and contribute to the development of healthier, higher-quality baked products.

## 1. Introduction

Wheat is one of the most important raw materials for preparing staple foods such as bread, biscuits, cookies, and pastries. It is a common component of human diets worldwide due to its versatility as a food item [1,2]. Today’s consumers demand foods that are not only flavorful, aromatic, and visually appealing but also promote health. Dietary fiber, for example, has gained popularity for its health benefits, including its role in preventing diabetes, cardiovascular diseases, hypertension, stroke, and cancer [3,4]. Baked products, as staple foods, are ideal sources of both total and soluble dietary fiber. By supplementing these foods with dietary fiber, we can increase fiber intake at a global scale [5,6].

Recent studies have explored the health benefits and bioactivities of cereal arabinoxylans. In cereal grains, arabinoxylans—polysaccharides found in the cell walls of the endosperm, aleurone layer, pericarp, and testa—are widely studied for their biochemical, structural, and gelling properties [7,8]. Ferulated arabinoxylans are soluble fibers present in major cereals, including wheat, barley, oats, rye, rice, sorghum, maize, and millet [8]. Increasing daily fiber consumption may reduce the risk of cardiovascular disease and type 2 diabetes, help to regulate blood sugar, and lower cholesterol [9].

Maize bran, a by-product of maize milling typically used as animal feed, is rich in dietary fiber and has potential applications in fiber-fortified foods [9]. The primary dietary fiber in maize bran, ferulated arabinoxylan, has a xylose backbone with arabinose side chains. In maize bran, ferulated arabinoxylan forms a major component of cell walls, consisting of (1→4)-linked β-d-xylopyranose units. The xylose units may be unsubstituted or substituted with one or two L-arabinofuranose units at the C-(O)-2 and/or C-(O)-3 positions [10,11]. FAXs may also contain other substituents, such as glucuronic acids, d-xylose, and hydroxycinnamic acids. Since ferulated arabinoxylan is a primary dietary fiber component in maize, it is increasingly regarded for its bioactivity and health benefits as a dietary fiber in cereals [12].

The distinct hydration behaviors of arabinoxylan fractions are associated with their chemical structure, substitution patterns, ferulic acid cross-links, and structural conformation. Even in small amounts, ferulated arabinoxylan significantly influences dough’s rheological properties and the quality characteristics of baked products [13]. Arabinoxylan has been shown to enhance viscosity, promote interfacial activity, and increase water absorption in dough, though its positive impact depends on the quantity and type of flour used [14,15]. Therefore, ferulated arabinoxylans can be used as food additives to stabilize and texturize food products.

In recent decades, the use of dietary fibers in baked goods has grown significantly, with various fiber sources, each with distinct chemical structures, being utilized [16]. In the food industry, particularly in baking, hydrocolloids are widely employed as dietary fiber ingredients. These hydrocolloid gums and gels are used to stabilize foams, emulsions, and suspensions, enhancing the sensory qualities, dough properties, and overall product quality of baked goods by modifying aqueous systems’ rheology and texture. Wheat flour containing arabinoxylan has been found to exhibit improved water absorption, pasting properties, extensibility, and gas pore stability [17,18,19].

Ferulated arabinoxylans (FAXs) are gaining attention as a dietary fiber due to their health benefits such as reduced gut infections, improved mineral absorption, and colon cancer suppression. Bioactive FAXs can therefore be incorporated into food products to improve nutritional value [20]. The purpose of this study is to extract ferulated arabinoxylans from maize bran and incorporate them into wheat flour. The current research reveals a gap in understanding how non-starch polysaccharides, particularly FAXs, influence flour mixture rheological and pasting properties. Rheological properties of the flour mixtures were studied through farinograph and mixograph analyses, while FAX and wheat flour blends were tested for pasting properties and micro-extensibility. The goal of this research is to increase knowledge about how non-starch polysaccharides can improve the functionality and gelation of flour and lead to healthier, higher-quality baked products.

## 2. Materials and Methods

### 2.1. Procurement of Raw Material

The current research work was conducted at the Whistler Center for Carbohydrate Research, Department of Food Science, Purdue University, West Lafayette, IN 47907, USA. Wheat flour was obtained from Ardent Mills (Denver, CO, USA). Maize bran was provided by Rafhan Maize Product Co Ltd., Faisalabad, Pakistan. Further milling was performed to obtain a <200 μm particle size of maize bran.

### 2.2. Extraction of Ferulated Arabinoxylans from Maize Bran

The FAXs were extracted from maize bran using the previous method of Hussain et al. [21] with some major modifications (using the alkaline solution method, instead of aqueous extraction; KOH 5%; and hydrolyzation for 5 h). Figure 1 shows the flow chart of the extraction method of FAXs from maize bran. Briefly, a 1000 g sample was dried and we defatted the sample using hexane (1:5 *w*/*v*). The sample was boiled in deionized water for 1 h for starch gelatinization, protein denaturation, and enzyme inactivation. The supernatant was discarded and then the sample was dried for 12 h at 60 °C. For hydrolysis, the sample was suspended in an alkaline solution, 5% KOH, and stirred for 5 h (150 rpm) at 35 °C. The slurry cooled down at room temperature for 1 h. Then, we centrifuged the material at 5000 rpm for 20 min at 4 °C. The supernatant was collected and precipitated in 70% *v*/*v* ethanol overnight. The precipitated material was recovered by the solvent exchange method (80% *v*/*v* acetone) and we obtained the extracted material. The extracted material was dried through a freeze dryer (−40 °C for 72 h).

### 2.3. Incorporation of FAXs in Wheat Flour

The extracted material FAXs were added into wheat flour with different concentrations, 0, 2, 4, 6, 8, and 10% in FAX0, FAX2, FAX4, FAX6, FAX8, and FAX10, respectively.

### 2.4. Moisture Content

The moisture contents of different FAX-incorporated wheat flour samples (FAX0, FAX2, FAX4, FAX6, FAX8, and FAX10) were determined by the AACC [22] method, 44–15.

### 2.5. Protein Content of FAX Extract

The protein contents of FAX-incorporated wheat flour samples (FAX0, FAX2, FAX4, FAX6, FAX8, and FAX10) were determined by a ThermoScientific Flashsmart Elemental Analyzer N/Protein instrument (Waltham, Massachusetts, USA). For sample preparation, the sample was dried and pressed into small tins to be weighed and inserted into the auto-sampler. Protein (%) was estimated by multiplying nitrogen (%) with a factor of 6.25.

### 2.6. Farinograph and Mixograph Study

The farinographic characteristics including the water absorption capacity, dough stability, Mixing Tolerance Index, dough development time, and farinograph quality numbers (mm) obtained from various blends of wheat flour with FAX treatments (FAX0, FAX2, FAX4, FAX6, FAX8, and FAX10) were analyzed using a farinograph (C.W. Brabender Farinograph, Model FA/R-2, Wood Dale, IL, USA) by following method number 54–21.02 of AACC-I [23].

FAX-blended wheat flour samples (FAX0, FAX2, FAX4, FAX6, FAX8, and FAX10) were tested on a mixograph at spring setting 11. Mixograms of dough obtained from various blends of FAXs and wheat flour treatments were analyzed according to AACC [22] method 54–40.02 to determine the mixing time and midline peak height, midline peak time, and midline peak integral.

### 2.7. Pasting Behavior of Flour Mixes

Pasting properties of different blends of wheat flour with FAX treatments (FAX0, FAX2, FAX4, FAX6, FAX8, and FAX10) were assessed by using a Rapid Visco Analyzer (RVA 4800, PerkinElmer, Perten instruments, Waltham, MA, USA) following AACCI [23] approved method 76–21.01. The flour quantity and water addition were estimated automatically by using RVA 4800 interfaces with PC and TCW3 (version 3.1) software. The profile of the tested samples includes the setback, hot paste viscosity, pasting temperature, final viscosity, breakdown, peak time, and peak viscosity of the FAX-incorporated wheat flours. TCW3 software was used to process the data.

### 2.8. Dough Extensibility

Dough extensibility of FAX-incorporated wheat flour treatments (FAX0, FAX2, FAX4, FAX6, FAX8, and FAX10) was estimated using a texture analyzer (Texture Technologies, TA-XT2i Analyzer, manufactured by Stable Micro Systems, United Kingdom). According to the Al-Saleh and Brennan [24] method, the extent of dough extensibility was assessed. Briefly, 0.5 g sodium chloride was added to a 25 g wheat flour sample, and water was added according to the farinograph water absorption results (Table 2). The sample was mixed with a lab-scale mixer (National Manufacturing Company, Lincoln, NE, USA) and we made dough according to the appropriate water absorption and development period as per faringraphic results (see Table 2). Samples were prepared according to the specifications of the Kieffer extensibility rig. After placing the dough balls on a dough clamp, they were allowed to rest at room temperature for 40 min. For assessing dough extensibility, a 2 kg load cell was calibrated by the texture analyzer. A one-by-one sample plate was loaded with the dough strips and placed in the sample holder. A tensile test was conducted on the Kieffer rig to determine extensibility, and data were collected using Exponent Connect.

### 2.9. Statistical Analysis

The results were analyzed statistically using Minitab 22 statistical software. The mean values and standard deviation of duplicate samples were estimated. A confidence level of 95% was used in the analysis of variance. The significance of differences between control and treated samples was evaluated using a test for least significant differences at *p* ≤ 0.05.

## 3. Results and Discussion

### 3.1. Extraction Yield of Ferulated Arabinoxylans

The current results confirmed that the extraction yield of FAXs from maize bran was 17.5 ± 0.5%. This yield is significantly higher than that reported in previous studies [21]. Based on our study, we achieved a higher yield by using a modified method of extraction, as opposed to aqueous extraction in previous studies. Thus, optimizing the extraction process can significantly increase the recovery of valuable compounds from maize bran.

### 3.2. Moisture and Protein Contents of FAX-Incorporated Wheat Flour

The moisture contents of FAX-incorporated wheat flour blends were 13 ± 0.2%, 10 ± 0.2%, 13 ± 0.3%, 14 ± 0.3%, 12 ± 0.6%, and 13 ± 0.4% in FAX0, FAX2, FAX4, FAX6, FAX8, and FAX10, respectively (Table 1). According to these findings, FAXs can influence moisture content, which is crucial for the shelf life and quality of wheat flour. There were no flour samples that contained moisture levels exceeding the maximum allowable limit of 14.5%, which is essential to ensure long-term storage stability [25]. Keeping moisture within this range is crucial to preventing microbial contamination, which can cause off-flavors, unpleasant odors, and mycotoxin development [26].

The protein contents of FAX and wheat flour blends were 12.86 ± 0.01%, 13 ± 0.04%, 13.15 ± 0.11%, 13.36 ± 0.16%, 13.65 ± 0.01%, and 14.05 ± 0.03% in FAX0, FAX2, FAX4, FAX6, FAX8, and FAX10, respectively. According to Table 1, the increasing trend of protein content of wheat flour blends with the increase in the FAX ratio was observed. There is a possibility that the increase in protein content is due to the proteins present in FAXs, which contribute to the nutritional profile of flour. It is well known that higher protein levels in dough result in better texture and quality in baked goods [27].

### 3.3. Farinograph and Mixograph Study of FAXs and Wheat Flour Blends

Wheat flour’s unique properties make it ideal for forming viscoelastic dough; however, adding fibers can alter the characteristics of dough made from composite flours. The farinograms generated by the farinograph test are shown in Appendix A. In farinograms, the first point where water addition intersects the curve top is termed the hydration time, while the point where the curve departs represents the departure time. Stability is defined by the time difference between hydration and departure. Additionally, peak time indicates the time from the initial water addition to the maximum dough consistency. The Mixing Tolerance Index (MTI) reflects the difference in Brabender Units (BU) between the peak and the curve top five minutes later. Hard wheat flour with an MTI of 30 BU or less is rated as very good to excellent in mixing tolerance, whereas an MTI above 50 BU suggests reduced tolerance and challenges during mechanical processing. Farinograph measurements of the dough quality of FAX-incorporated wheat flour blends are presented in Table 2. Our findings indicate that wheat flour (FAX0) had a water absorption of 54.9 ± 0.00%, while FAX-enriched composite doughs ranged from 56.1 ± 1.06% to 60.5 ± 0.14%. Water absorption increased significantly (*p* ≤ 0.05) with higher FAX content, comparing directly with the amount of FAXs in the wheat flour blends. As observed in recent studies by Pietiäinen et al. [20], wheat doughs containing arabinoxylans displayed increased water absorption. Flour needs a specific amount of water for rheological testing and product quality evaluations to ensure optimal dough handling. The increased water absorption can be attributed to the high arabinose and xylose content of FAXs, which absorb more water due to their chemical structure. For instance, arabinoxylans have been shown to absorb water 4 to 10 times their weight [28]. This enhanced water absorption can be attributed to the hygroscopic nature of FAXs, which integrate well into the dough matrix, elevating its water-holding capacity. Various factors, such as the fiber source, structure, extraction method, porosity, and particle size, influence water absorption in these blends. Hydrophilic interactions and hydrogen bonds formed between protein and starch molecules also contribute to hydration [29]. Our study found that dough development times ranged from 1.7 to 24.7 ± 0.07 min. Notably, the 2% FAX treatment (FAX2) showed an extended dough development time of 24.7 ± 0.07 min compared to other treatments. Generally, dough development time decreases as FAX concentration rises due to AX’s water-binding properties, which delay gluten development and reduce mixing time [30]. This unique behavior suggests that low concentrations of FAXs may interact differently with gluten networks, enhancing dough cohesiveness and requiring additional development time. Farinograph parameters like MTI, stability, and the quality number indicate dough strength and mixing tolerance. MTI values ranged from 5.5 ± 0.7 to 87.5 ± 0.7 BU, while stability ranged from 1.8 to 41.8 ± 2.26 min. Increasing FAX levels resulted in decreased dough strength and mixing tolerance. A significant drop in peak time was observed at the 2% FAX addition level. These parameters—peak time, stability, MTI, and quality number—reflect dough strength and mixing tolerance, largely dependent on gluten protein quality. With each incremental FAX addition (4–10%), stability decreased by more than half. Higher FAX concentrations result in a loss of consistency and mixing tolerance due to fiber and gluten protein interactions. This finding aligns with studies showing that dietary fiber in composite flours can reduce dough strength [31], although increased water absorption can negatively impact dough strength.

The mixograph, which records dough’s rheological properties during mixing, provides information on key parameters such as time to maximum curve height (minimum mobility), maximum curve height, and ascending and descending curve angles [32]. The mixograms for various FAX and wheat flour blends are shown in Appendix A. The mixograph analysis revealed that FAX-incorporated wheat flours displayed distinct midline peak time (MPT), midline peak height (MPH), and midline peak integral (MPI) characteristics. Dough mixing occurs as flour and water combine until gluten forms from the interaction of dispersed and hydrated gluten-forming proteins [33].

A significant reduction in peak time was observed with FAX addition (*p* ≤ 0.05), although the 2% FAX treatment (FAX2) demonstrated a higher mixing time (14.4 ± 3.3 min) compared to other treatments. The control (FAX0) showed a higher mixing time of 5.55 ± 0.9 min, while the peak times for FAX4 (2.85 ± 0.5), FAX6 (1.35 ± 0.1), FAX8 (1.3 ± 0.00), and FAX10 (1.15 ± 0.00) (Table 2) were different. Mixograph peak height, which indicates dough strength, decreased gradually as FAX levels increased. The FAX0 control exhibited the significantly highest peak height at 44.15 ± 0.6%, while peak heights for FAX2, FAX4, FAX6, FAX8, and FAX10 were 36.65 ± 0.1%, 33.95 ± 0.3%, 33.65 ± 0.3%, 34.3 ± 0.6%, and 33.4 ± 0.3%, respectively. The higher peak height for FAX2 suggests greater dough stiffness due to increased flour protein content [34].

In various mixograms, the curve height generally correlates with protein content at optimal hydration. The curve shape, particularly on the right side, reflects optimal water absorption, while curves with wild swings before the peak may indicate the need for more water. The mixograph curve provides a visual estimate of optimally developed dough, as indicated by MPH and MPT. The curve width at the peak reflects dough elasticity, while the breakdown angle post-peak indicates mixing tolerance. A broader curve width after overmixing corresponds to more elastic dough, and multiplying this width by the mixing time (e.g., 50%) reflects the degree of mixing tolerance. Further mixing typically results in curve decline, marking the onset of dough breakdown.

### 3.4. Pasting Properties of FAX-Incorporated Wheat Flour Blends

A Rapid Visco Analyzer (RVA) was used to record the apparent viscosity of aqueous wheat flour suspensions to generate pasting curves for different wheat flours (whole wheat flours, refined wheat flours, and composite wheat flour blends). The pasting curves for FAX-incorporated wheat flour blends are shown in Figure 2. Pasting properties reflect changes in food when heat is applied in the presence of water, which can influence the texture, sensory qualities, digestibility, and overall end-use of the food product. Various attributes of each flour in the blend affect starch pasting properties, and RVA measurements show that fiber addition impacts pasting profiles across all flour types.

In this study, peak viscosity decreased significantly (*p* < 0.05) from 2384.5 ± 27.5 to 1163 ± 100.4 mPa⋅s as FAX content increased. This reduction in peak viscosity may be attributed to the lower starch content and the interactions among fiber, protein, and starch within the blends. Peak viscosity is linked to the water-binding capacity of starch, which occurs at equilibrium between swelling (increasing viscosity) and rupture (decreasing viscosity). Therefore, FAX composite wheat flour, with its high dietary fiber and low starch content, exhibited reduced viscosity.

FAXs significantly reduced hot paste viscosity (HPV) when blended with wheat flour at all inclusion levels (*p* > 0.05). HPV values ranged from 1332.5 ± 33.2 mPa⋅s to 914 ± 74.9 mPa⋅s, indicating that hot paste viscosity decreased significantly as FAX content in wheat flour increased. A lower HPV correlates with decreased breakdown viscosity, suggesting the blends’ increased resistance to breakdown under heating and shearing. Setback values, ranging from 1070 ± 24 to 1909.5 ± 21.9 mPa.s, relate to the retrogradation and staling properties of starch-based foods. Compared with the wheat flour control, FAXs reduced peak time, indicating an altered pasting profile. Lower breakdown viscosity also suggests enhanced heat and shear tolerance during cooking, as supported by Adebowale et al. [35].

Peak time reflects the time required for complete starch gelatinization, a critical indicator of the paste’s cooking characteristics. While there was no significant difference in pasting temperature between 100% wheat flour (FAX0) and blends with 2–8% FAXs (FAX2, FAX4, FAX6, and FAX8), pasting temperatures decreased significantly (*p* < 0.05) at a 10% FAX concentration.

In this study, the wheat flour was much finer than the FAX powder, which had a larger particle size, influencing the flour’s hydration rate. Since FAX powder, derived from maize bran, was freeze-dried and ground into a powder, its particle size impacts the pasting behavior under varying processing conditions. Additionally, amylase enzymes in wheat flour hydrolyze starch, affecting paste viscosity [36]. The pasting properties of flour are influenced by several complex factors, and due to the compositional complexity of the flour blends, fewer significant (*p* > 0.05) differences may be observed in their pasting profiles. Table 3 summarizes the effects of FAX addition on the pasting properties of wheat flour.

### 3.5. Micro-Extensibility

Dough extensibility measures the total horizontal distance the dough curve travels from the start of stretching until it breaks. In dough, the maximum force (N) represents resistance to extension, while the distance to break (mm) reflects extensibility. With fiber addition, dough extensibility decreased, and resistance to extension increased compared to the control, indicating that higher fiber content enhances dough firmness and weakens its extensibility. This results in FAX0 (43.18 mm) and FAX2 (43.18 mm) exhibiting higher extensibility than FAX4 (12.5 mm), FAX6 (18.2 mm), FAX8 (16.39 mm), and FAX10 (17.25 mm) (Figure 3). Since dough extensibility is linked to dough strength, these results suggest that FAX concentrations above 2% significantly weaken the dough, with the weakening effect intensifying as FAX content increases. Previous studies indicate that fiber incorporation limits gas retention in the gluten network, hindering gluten agglomeration and thereby reducing dough extensibility. This is attributed to the disruption of the starch–gluten matrix, which restricts gas retention in the gluten network. Additionally, these findings align with evidence showing that fiber-enriched blends more effectively disrupt gluten networks than control blends due to their higher water-holding capacity (WHC), water absorption, and viscosity [37].

Dough area values decreased with the addition of AX, indicating that less energy was needed to deform the dough structure. Increased extensibility corresponded to decreased extension resistance, resulting in a significant decrease (*p* < 0.05) in the extensibility-to-resistance ratio. In this study, FAX4 exhibited high resistance (97.38 F) compared to FAX0 (36.62 F) and FAX2 (36.62 F). Furthermore, FAX6, FAX8, and FAX10 showed extensibility resistances of 48.77 F, 42.7 F, and 39.16 F, respectively (Figure 3). These results demonstrate the positive effect of AX on the gluten network, as gluten complexes enhance the dough’s extensibility. The AX–starch–gluten interaction promotes macromolecular connectivity, reducing the stretching force required and resulting in dough prone to extensional deformation. Consequently, dough enriched with low-dose AX exhibits reduced stickiness, greater extensibility, and improved strain resistance.

## 4. Conclusions

The incorporation of ferulated arabinoxylans (FAXs) into wheat flour at varying concentrations significantly impacts the rheological and pasting properties of the resulting FAX–wheat flour blends. The extent of these changes is largely influenced by the biochemical properties of the added bioactive compounds. Our findings show that farinograph water absorption for wheat and FAX blends ranged from 54.9% to 60.5%, with water absorption increasing significantly (*p* ≤ 0.05) as FAX content increased.

Furthermore, as the FAX concentration increased, dough development time decreased significantly (*p* ≤ 0.05), with the exception of the 2% FAX treatment (FAX2), which displayed a notably longer development time of 24.7 min. All treatments displayed significant variations in stability and Mixing Tolerance Index (MTI) values (*p* ≤ 0.05), with stability generally decreasing as FAX concentration increased, indicating an inverse relationship between fiber addition and dough stability under extended mixing.

Pasting properties also demonstrated notable effects due to FAX incorporation. Peak viscosity and hot paste viscosity both decreased significantly (*p* < 0.05) as FAX content increased, which is indicative of the lower starch content and the interaction between starch, protein, and fiber in the blends. This interaction may reduce starch’s ability to swell fully, leading to a decrease in hot paste viscosity and, consequently, a lower breakdown viscosity, suggesting enhanced resilience of the dough during heating and shearing.

Micro-extensibility tests revealed that FAX0 and FAX2 blends exhibited the greatest extensibility (43.18 mm each), while higher FAX concentrations (FAX4, FAX6, FAX8, and FAX10) led to decreased extensibility. Specifically, the FAX4 treatment demonstrated the highest resistance (97.38 F), suggesting that this concentration reinforces the dough’s structural integrity more than higher FAX levels, which might lead to weaker and less cohesive gluten networks. These variations reflect FAX’s potential to modify dough characteristics, making it possible to tailor dough performance based on FAX concentration to meet specific functional and textural requirements in baking applications.

In conclusion, FAX addition profoundly influences the rheological and pasting properties of wheat flour. The results underscore the value of FAXs as a functional ingredient with bioactive properties that could be effectively leveraged to enhance the nutritional profile and modify the structural characteristics of baked products. By understanding how different concentrations of FAX affect dough properties, the baking industry can better integrate FAXs into formulations to create products with desirable texture, enhanced fiber content, and improved processing qualities. These findings contribute to the growing body of research on fiber-enriched baked goods and open pathways for further studies on optimizing FAXs for specific food applications.

## Figures and Tables

**Figure 1 foods-13-03414-f001:**
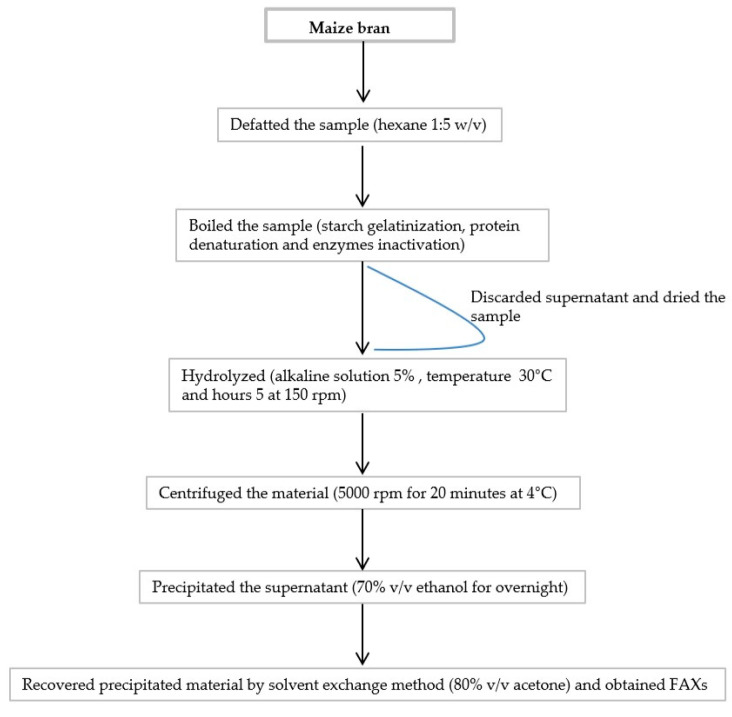
Flow chart of extraction method of ferulated arabinoxylans from maize bran: FAXs, ferulated arabinoxylans; rpm, revolutions per minute; *w*/*v*, weight/volume; *v*/*v*, volume/volume.

**Figure 2 foods-13-03414-f002:**
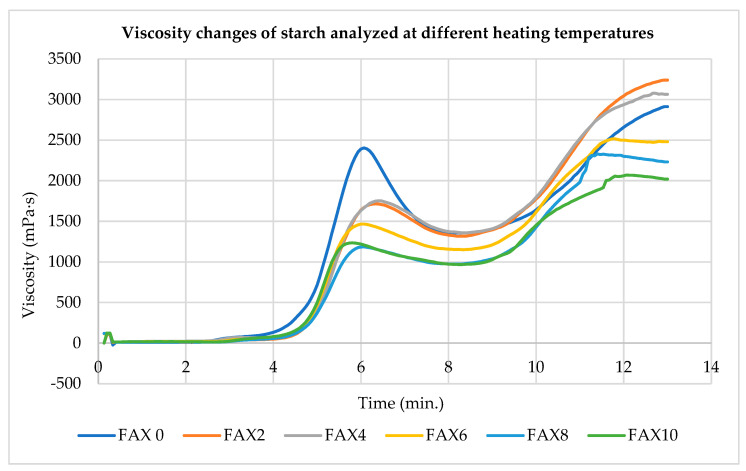
Effect of ferulated arabinoxylan addition on viscosity changes in starch at different heating temperatures. FAX0: control (100% wheat flour); FAX2: 2% ferulated arabinoxylans + 98% wheat flour; FAX4: 4% ferulated arabinoxylans + 96% wheat flour; FAX6: 6% ferulated arabinoxylans + 94% wheat flour; FAX8: 8% ferulated arabinoxylans + 92% wheat flour; FAX10: 10% ferulated arabinoxylans + 90% wheat flour; mPa⋅s: millipascal-second.

**Figure 3 foods-13-03414-f003:**
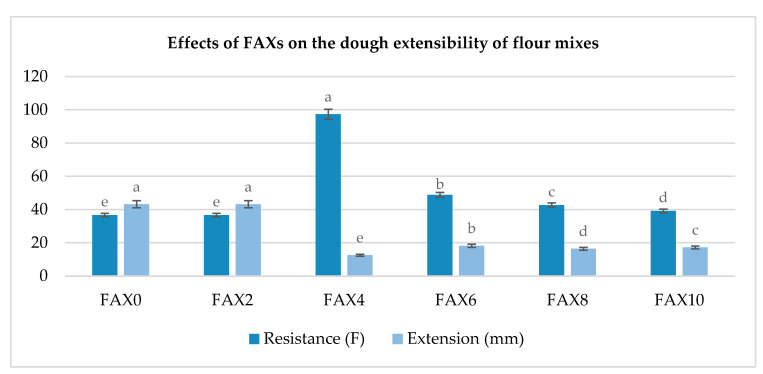
Effect of ferulated arabinoxylan addition on micro-extensibility of wheat flour blends. FAXs: ferulated arabinoxylans; FAX0: control (100% wheat flour); FAX2: 2% ferulated arabinoxylans + 98% wheat flour; FAX4: 4% ferulated arabinoxylans + 96% wheat flour; FAX6: 6% ferulated arabinoxylans + 94% wheat flour; FAX8: 8% ferulated arabinoxylans + 92% wheat flour; FAX10: 10% ferulated arabinoxylans + 90% wheat flour. Values followed by different letters (a–e) above same-color bars are significantly different at *p* ˂ 0.05.

**Table 1 foods-13-03414-t001:** Moisture and protein contents of ferulated arabinoxylans and wheat flour blends.

Treatments	Moisture Content (%)	Protein Content (%)
FAX0	13 ± 0.2 b	12.86 ± 0.01 f
FAX2	10 ± 0.2 d	13 ± 0.04 e
FAX4	13 ± 0.3 b	13.15 ± 0.11 d
FAX6	14 ± 0.3 a	13.36 ± 0.16 c
FAX8	12 ± 0.6 c	13.65 ± 0.01 b
FAX10	13 ± 0.4 b	14.05 ± 0.03 a

FAX0: control (100% wheat flour); FAX2: 2% ferulated arabinoxylans + 98% wheat flour; FAX4: 4% ferulated arabinoxylans + 96% wheat flour; FAX6: 6% ferulated arabinoxylans + 94% wheat flour; FAX8: 8% ferulated arabinoxylans + 92% wheat flour; FAX10: 10% ferulated arabinoxylans + 90% wheat flour. Different letters (a–f) within the column indicate that the interactions of treatments are significantly different (*p* ≤ 0.05). The same letters show non-significant differences from each other. Results are expressed as the mean value ± standard deviation (n = 3).

**Table 2 foods-13-03414-t002:** Results of Farinograph and Mixograph study of ferulated arabinoxylans and wheat flour blends.

Treatments		FAX0	FAX2	FAX4	FAX6	FAX8	FAX10
Farinograph	Flour Moisture (%)	15.5 ± 0.17 a	15.7 ± 0.09 a	15.5 ± 0.27 a	15.7 ± 0.2 a	15.6 ± 0.19 a	15.6 ± 0.19 a
	Absorption % (14% m.b.)	54.9 ± 0.00 e	56.1 ± 1.06 d	57.2 ± 0.21 c	57.4 ± 0.14 c	58.1 ± 0.00 b	60.5 ± 0.14 a
	Development Time (min)	7.7 ± 0.07 b	24.7 ± 0.07 a	3.6 ± 0.49 c	2.5 ± 0.35 d	1.7 ± 0.07 e	1.7 ± 0.00 e
	Stability (min)	14.3 ± 0.42 b	41.8 ± 2.26 a	5.2 ± 0.28 c	4 ± 0.02 d	1.9 ± 0.14 e	1.8 ± 0.00 e
	M.T.I. (BU)	27 ± 1.4 d	5.5 ± 0.7 e	55.5 ± 2.1 c	58.5 ± 0.7 c	78 ± 1.4 b	87.5 ± 0.7 a
	Time to Breakdown (min)	13.2 ± 0.1 b	46.46 ± 3 a	5.2 ± 0.1 c	4 ± 0.3 d	1.9 ± 0.1 e	1.8 ± 0.00 e
	Farinograph Quality Number (mm)	132 ± 1.4 b	464.5 ± 30.4 a	65 ± 1.4 c	52 ± 2.8 d	31 ± 1.4 e	28.5 ± 0.7 e
Mixograph	Absorption % (14% m.b.)	58 ± 0.00 b	55.6 ± 0.00 d	57.2 ± 0.00 c	57.3 ± 0.00 c	57.7 ± 0.6 c	60.4 ± 0.00 a
	Midline Peak Time (min)	5.55 ± 0.9 b	14.4 ± 3.3 a	2.85 ± 0.5 c	1.35 ± 0.1 d	1.3 ± 0.00 d	1.15 ± 0.00 d
	Midline Peak Height (%)	44.15 ± 0.6 a	36.65 ± 0.1 b	33.95 ± 0.3 d	33.65 ± 0.3 d	34.3 ± 0.6 c	33.4 ± 0.3 d
	Midline Peak Integral (%TQ MIN)	206.5 ± 35.5 b	485.35 ± 109 a	86.1 ± 16.7 c	37.8 ± 5 d	35.95 ± 0.8 d	31.1 ± 0.7 e

FAX0: control (100% wheat flour); FAX2: 2% ferulated arabinoxylans + 98% wheat flour; FAX4: 4% ferulated arabinoxylans + 96% wheat flour; FAX6: 6% ferulated arabinoxylans + 94% wheat flour; FAX8: 8% ferulated arabinoxylans + 92% wheat flour; FAX10: 10% ferulated arabinoxylans + 90% wheat flour; m.b: moisture basis; M.T.I: Mixing Tolerance Index; BU: Brabender Units; TQ: torque. Different letters (a–e) within the row indicate that the interactions of treatments are significantly different (*p* ≤ 0.05). The same letters show non-significant differences from each other. Results are expressed as the mean value ± standard deviation (n = 2).

**Table 3 foods-13-03414-t003:** Effect of ferulated arabinoxylans on pasting properties of wheat flour.

Treatments	Peak Viscosity mPa.s	Hot Paste Viscosity mPa.s	Breakdown mPa.s	Final Viscosity mPa.s	Setback mPa.s	Peak Time min	Pasting Temp °C
FAX0	2384.5 ± 27.5 a	1332.5 ± 33.2 a	1052 ± 5.6 a	2888 ± 35.3 b	1555.5 ± 2.1 c	6.03 ± 0.04 a	88.7 ± 0.03 b
FAX2	1677.5 ± 53 b	1282.5 ± 45.9 b	395 ± 7 b	3192 ± 67.8 a	1909.5 ± 21.9 a	6.36 ± 0.04 a	92.8 ± 0 a
FAX4	1692.5 ± 85.5 b	1346 ± 16.9 a	346.5 ± 68.5 b	3074.5 ± 14.8 a	1728.5 ± 31.8 b	6.4 ± 0 a	93.6 ± 0.03 a
FAX6	1455 ± 15.5 c	1155.5 ± 7.7 c	299.5 ± 23.3 c	2553 ± 103.2 c	1397.5 ± 95.4 d	6.1 ± 0.14 a	92.9 ± 0 a
FAX8	1183 ± 2.8 d	965 ± 11.3 d	218 ± 8.4 d	2250.5 ± 27.5 d	1285.5 ± 38.8 d	6.1 ± 0.04 a	93.2 ± 0.7 a
FAX10	1163 ± 100.4 d	914 ± 74.9 d	249 ± 25.4 c	1984 ± 50.9 e	1070 ± 24 e	5.8 ± 0.04 b	92.3 ± 0.6 a

FAX0: control (100% wheat flour); FAX2: 2% ferulated arabinoxylans + 98% wheat flour; FAX4: 4% ferulated arabinoxylans + 96% wheat flour; FAX6: 6% ferulated arabinoxylans + 94% wheat flour; FAX8: 8% ferulated arabinoxylans + 92% wheat flour; FAX10: 10% ferulated arabinoxylans + 90% wheat flour. Different letters (a–e) within the column indicate that the interactions of treatments are significantly different (*p* ≤ 0.05). The same letters show non-significant differences from each other. Results are expressed as the mean value ± standard deviation (n = 2). mPa⋅s: millipascal-second.

## Data Availability

The data presented in this study are available on request from the corresponding author. The data are not publicly available due to privacy restrictions.

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
