# Peer review of "Impact of Ferulated Arabinoxylans from Maize Bran on Farinograph and Pasting Properties of Wheat Flour Blends"

_foods, 2024, doi:10.3390/foods13213414_

Round 1

Reviewer 1 Report

Comments and Suggestions for Authors

Review on manuscript: foods-3285978

Impact of Ferulated Arabinoxylans from Maize Bran on the Rheological and Pasting Properties of Wheat Flour Blends by Muzzamal Hussain, Senay Simsek submitted to Foods

In the manuscript submitted for review, the authors studied the effect of ferulated arabinoxylans from maize bran on the rheological properties of wheat flour.

In my opinion, the topic taken up by the authors is not new, and the authors did not indicate what new knowledge their results contribute, moreover, the manuscript was prepared carelessly and requires many corrections.

Detailed recommendation:

lines 68-73 – the introduction should end with a clearly formulated research objective resulting from the literature review, and not with a summary of the methods used, at this point, the authors should also indicate what new information their research results bring to the knowledge they already have,

line 85 – if the original method has been significantly modified, its description should be extended,

Figure 1 – temperature and drying time should be given, degree sign is missing, instead of rpm centrifugal force should be given, is the use of acetone justified here? how is the acetone residue removed?

line 100 – origin country should be added, if nitrogen content was determined, what protein multiplier was used?

lines 116-117 – RVA devices are not manufactured by PerkinEmer, and Perten Instruments comes from Sweden,

lines 119-120 – the order of the given parameters should be logical: gelatinization temperature, peak viscosity, peak time, hot paste viscosity ..., why do the authors randomly use capital letters?

lines 125-126 – the TA-XT2i is manufactured by Stable Micro Systems, United Kingdom,

line 129 – origin country should be added,

line 141 – what specific post hoc test was used?

lines 150-151, 157-158, 223, 226, 296-297 – authors should strictly limit the repetition of numerical data available in tables or figures,

Table 1 – moisture content should be reported with greater accuracy, data should be reported with the same accuracy,

lines 174-184 – theoretical description of device operation and parameter definitions are not necessary here,

line 188 – the value of the linear correlation coefficient could be given,

Table 2 – the format does not comply with the requirements,

lines 252, 259, 262, Figure 2, Table 3 – viscosity values ​​should be expressed in SI units,

Conclusion – these should be real conclusions and not a repetition of the results obtained,

References – authors inconsistently provide full titles of journals or their abbreviations - this should be standardized as required.

Author Response

18 October 2024

Dear Referee,

We would like to thank the referee for the close reading and for the proper suggestions. We hope that we provide all the answers to the reviewer’s comments.

Thank you very much for the recommendations to publish our paper entitled “Impact of Ferulated Arabinoxylans from Maize Bran on the Rheological and Pasting Properties of Wheat Flour Blends”.

The present version of the paper has been revised according to the reviewer’s suggestions.          

We uploaded the corrected version of the article for which we used the track changes for the addition text.

Response to Reviewer 1 Comments

Detailed recommendations:

lines 68-73 – the introduction should end with a clearly formulated research objective resulting from the literature review, and not with a summary of the methods used, at this point, the authors should also indicate what new information their research results bring to the knowledge they already have,

Response: We would like to thank to the referee for his/her close reading of our manuscript data. We agree with this comment. The suggestion has now been incorporated at same place (line 70-71). The purpose of this study is to extract ferulated arabinoxylans from maize bran and incorporate them into wheat flour. The current research reveals a gap in understanding how non-starch polysaccharides, particularly FAX, influence flour mixture rheological and pasting properties. Rheological properties of the flour mixtures were studied through farinograph and mixograph analyses, while FAX and wheat flour blends were tested for pasting properties and micro-extensibility. The goal of this research is to increase knowledge about how non-starch polysaccharides can improve the functionality and gelation of flour and lead to healthier, higher-quality baked products.

line 85 – if the original method has been significantly modified, its description should be extended,

Response: Thanks for your valuable comment. The suggestion has now been incorporated and updated the extarction method (line 88-94) as;

Briefly, 1000g sample was dried and defatted the sample using hexane (1:5 w/v). The sample was boiled in deionized water for 1 hour to starch gelatinization, protein denaturation and enzymes inactivation. The supernatant was discarded and then the sample was dried for 12 hours at 60°C. For hydrolysis, sample suspended in alkaline solution 5% KOH and stirred for 5 hours (150 rpm) at 35°C. The slurry cooled down at room temperature for 1 hour. Then, centrifuge the material at 5000 rpm for 20 minutes at 4°C. Supernatant was collected and precipitated in 70% v/v ethanol for overnight. The precipitated material was recovered by solvent exchange method (80% v/v acetone) and obtained extracted material.

Figure 1 – temperature and drying time should be given, degree sign is missing, instead of rpm centrifugal force should be given, is the use of acetone justified here? how is the acetone residue removed?

Response: Thanks for your valuable suggestions. The drying time  and temperature have now been added in line 94-95 as „The extracted material was dried through freeze dryer (-40 °C for 72 hours”. Degrre sign has been added at same place.

For residue removal, we gently heat the mixture to evaporate the acetone (under a fume hood). Acetone has a relatively low boiling point (56 °C), so easily evaporated from extracted material. Additionally, freeze-drying method is also useful to remove acetone residues from the remaining extract.

line 100 – origin country should be added, if nitrogen content was determined, what protein multiplier was used?

Response: The origin country has been added (Waltham, Massachusetts, USA).  Protein (%) was estimated by multiplying nitrogen (%) with a factor of 6.25.

lines 116-117 – RVA devices are not manufactured by PerkinEmer, and Perten Instruments comes from Sweden,

Response: The source country mentioned in text is already corrected (RVA 4800, PerkinElmer, Perten instruments, USA). Location; 209 Otley St, Leighton, IA 50143, US

lines 119-120 – the order of the given parameters should be logical: gelatinization temperature, peak viscosity, peak time, hot paste viscosity ..., why do the authors randomly use capital letters?

Response: Thanks for your valuable suggestions. The suggestion has now been done at sampe place.

lines 125-126 – the TA-XT2i is manufactured by Stable Micro Systems, United Kingdom,

Response: Thanks for your valuable suggestions. Your suggestion has been incorporated accordingly.

line 129 – origin country should be added,

Response: The country has now been added NE, USA

line 141 – what specific post hoc test was used?

Response: We used Tukey's HSD in current analysis

lines 150-151, 157-158, 223, 226, 296-297 – authors should strictly limit the repetition of numerical data available in tables or figures,

Response: Thanks for your valuable suggestions. The stetements have been modified at same places. The mentioned sentences have been rephrased.

Table 1 – moisture content should be reported with greater accuracy, data should be reported with the same accuracy,

Response: The moisture content values reported as according to results data with great accuracy.

lines 174-184 – theoretical description of device operation and parameter definitions are not necessary here,

Response: Dear reviewer, we removed the description section line 174-177 (A farinograph, a type of dough mixer, records and measures the torque exerted by the mixer blades during mixing. This tool is essential for estimating flour water absorption, relative mixing time, dough stability under overmixing, and rheological properties). However, we can not remove the farinograms discussion line 177-184 because this data provided the information regarding supplementary file 1 (Farinographs).

line 188 – the value of the linear correlation coefficient could be given,

Response: We repharse the word correlation with comparing for better understading

Table 2 – the format does not comply with the requirements,

Response: Thanks for your valuable suggestion. Table 2 has now been reformatted according to journal style.

lines 252, 259, 262, Figure 2, Table 3 – viscosity values ​​should be expressed in SI units,

Response: Thanks for your valuable suggestion. The viscosity values ​​have now been expressed in SI unit mPas in Table 3, figure 2 and manuscript text.

Conclusion – these should be real conclusions and not a repetition of the results obtained,

Response: Thanks for your valuable suggestion. The statments regarding results repetiion have now been removed and added into results and discussion section.

References – authors inconsistently provide full titles of journals or their abbreviations - this should be standardized as required.

Response: The reference list has now been improved according to journal requirment.

*Finally, the authors would like to thank reviewers for their appreciation and for all the suggestions because these helped us to correct our paper and to improve it.

Sincerely,

Muzzamal Hussain et al.

Reviewer 2 Report

Comments and Suggestions for Authors

The "recycling" of waste or byproducts from the technological processing of plant material is a very important topic, especially if it deals with obtaining products suitable for human consumption. Therefore, the manuscript "Impact of Ferulated Arabinoxylans from Maize Bran on the Rheological and Pasting Properties of Wheat Flour Blends" is very interesting.

The introduction is quite nicely written, however there is a lack of information on whether there are already similar products on the market and why the authors decided on ferulated arabinoxylans from maize bran.

Also, it is necessary to emphasize whether this product, i.e. incorporation of ferulated arabinoxylans into wheat flour, has a perspective for application on a global level!

Author Response

18 October 2024

Dear Referee,

We would like to thank the referee for the close reading and for the proper suggestions. We hope that we provide all the answers to the reviewer’s comments.

Thank you very much for the recommendations to publish our paper entitled “Impact of Ferulated Arabinoxylans from Maize Bran on the Rheological and Pasting Properties of Wheat Flour Blends”.

The present version of the paper has been revised according to the reviewer’s suggestions. 

We uploaded the corrected version of the article for which we used the track changes for the addition text.

Response to Reviewer 2 Comments

The "recycling" of waste or byproducts from the technological processing of plant material is a very important topic, especially if it deals with obtaining products suitable for human consumption. Therefore, the manuscript "Impact of Ferulated Arabinoxylans from Maize Bran on the Rheological and Pasting Properties of Wheat Flour Blends" is very interesting.

Response: The referee was kind enough to read and review our manuscript closely. We thank him/her for his/her thoughtful remarks. In the future, this research will lead to the development of healthier, higher-quality baked goods.

The introduction is quite nicely written, however there is a lack of information on whether there are already similar products on the market and why the authors decided on ferulated arabinoxylans from maize bran.

Response: Regarding the introduction, we appreciate your positive comments. In response to your concerns, we improved the introduction by including utilize ferulated arabinoxylans and their applications in the market (58-59). We also clarified our rationale for selecting ferulated arabinoxylans from maize bran (51-52). In addition, previous literature showed that maize bran is a good source of ferulated arabinoxylans as compared to other cereal bran.

Also, it is necessary to emphasize whether this product, i.e. incorporation of ferulated arabinoxylans into wheat flour, has a perspective for application on a global level!

Response: Dear refree, we added the following statment in introduction section to clear this perspective. The current research reveals a gap in understanding how non-starch polysaccharides, particularly ferulated arabinoxylans, influence flour mixture rheological and pasting properties.

*Finally, the authors would like to thank the reviewers for their appreciation and suggestions because these helped us correct our paper and optimize it.

Sincerely,

Muzzamal Hussain et al.

Reviewer 3 Report

Comments and Suggestions for Authors

The authors extracted ferulated arabinoxylans from maize bran and mixed it with wheat flour and then tested the pasting, farinograph and extensile properties of the mixed flour. Adding some new and nutritious ingredients into wheat flour is a new trend in these years. However, the manuscript have several flaws that I may concern.

1.     The manuscript didn’t reflect the careless of the authors, the abstract in the submitting system is not complete. Figure 1 is not clear enough.

2.     Title: I suggest the authors replace rheological with farinograph is better.

3.     Keyword: too much, 3-5 are enough.

4.     Introduction: you mentioned Ferulated arabinoxylan in line 41 and then Arabinoxylan in line 66. Is there any difference between them two, you should clarify it.

5.     Figure 1: you should add the picture of your extracted FAXs, and the most important question is that have you characterized the extraction. How do you know the purity of your extraction. I didn’t see the details in this manuscript and any related reference.

6.     Line 144: you mentioned the yield of FAXs is 17.5%, how do you calculated it and the author should provide the proof that the extraction is FAXs.

7.     Line 92-93: where did you get the wheat flour, what’s the brand?

8.     Table 2: the development time and stability of FAX2 is abnormal I think the authors should double check it. And I checked the supplementary materials they just replicate the test twice, I don’t think it’s very convincing.

9.     Figure 3: what’s the unit of the ordinate?

10.  Conclusion: it’s too long, make it short and clear. 

Author Response

18 October 2024

Dear Referee,

We would like to thank the referee for the close reading and for the proper suggestions. We hope that we provide all the answers to the reviewer’s comments.

Thank you very much for the recommendations to publish our paper entitled “Impact of Ferulated Arabinoxylans from Maize Bran on the Rheological and Pasting Properties of Wheat Flour Blends”.

The present version of the paper has been revised according to the reviewer’s suggestions. 

We uploaded the corrected version of the article for which we used the track changes for the addition text.

Response to Reviewer 3 Comments

The authors extracted ferulated arabinoxylans from maize bran and mixed it with wheat flour and then tested the pasting, farinograph and extensile properties of the mixed flour. Adding some new and nutritious ingredients into wheat flour is a new trend in these years. However, the manuscript have several flaws that I may concern.

  1. The manuscript didn’t reflect the careless of the authors, the abstract in the submitting system is not complete. Figure 1 is not clear enough.

Response: Thanks for your valuable suggestions. The abstract is not uploaded due to system error. The complete abstarct has now been added in main manuscript file. Figure 1 has now been improved.

  1. Title: I suggest the authors replace rheological with farinograph is better.

Response: Thanks for your valuable suggestions. We modified the title as per your suggestion; Impact of Ferulated Arabinoxylans from Maize Bran on the farinograph and Pasting Properties of Wheat Flour Blends

  1. Keyword: too much, 3-5 are enough.

Response: Thanks for your valuable suggestion. Keywords have now been modified.

  1. Introduction: you mentioned Ferulated arabinoxylan in line 41 and then Arabinoxylan in line 66. Is there any difference between them two, you should clarify it.

Response: Thanks for your valuable suggestions. Ferulated arabinoxylan added in line 66.

  1. Figure 1: you should add the picture of your extracted FAXs, and the most important question is that have you characterized the extraction. How do you know the purity of your extraction. I didn’t see the details in this manuscript and any related reference.

Response: We comprehensivelly characterized the extracted material in a previous published work (https://www.mdpi.com/2304-8158/11/21/3374). In addition, we already added the original picture of extracted material in graphical abstract.

  1. Line 144: you mentioned the yield of FAXs is 17.5%, how do you calculated it and the author should provide the proof that the extraction is FAXs.

Response: Our previously published paper (https://www.mdpi.com/2304-8158/11/21/3374) provided the biochemiacl, structural characterization of FAXs. extract. In current study, we calculated the yield of FAXs using following formula;

The yield of FAXs was calculated from the following Eq. (1):

Yield (%) = W1/W0×100 (1)

Where W1 is the weight of dried FAX (g), W0 is the weight of dry bran (g).

  1. Line 92-93: where did you get the wheat flour, what’s the brand?

Response: Information regarding wheat flour is mentioned in “Procurement of raw material” section as; Wheat flour was obtained from Ardent Mills (Denver, CO, USA).

  1. Table 2: the development time and stability of FAX2 is abnormal I think the authors should double check it. And I checked the supplementary materials they just replicate the test twice, I don’t think it’s very convincing.

Response: The results were the same when we double-checked them. As a result, we mentioned the results based on the farinograph.

  1. Figure 3: what’s the unit of the ordinate?

Response: F in mm

  1. Conclusion: it’s too long, make it short and clear. 

Response: Thanks for your valuable suggestion. We have now improved and shoerten the conclusion section.

*Finally, the authors would like to thank reviewers for their appreciation and for all the suggestions because these helped us to correct our paper and to optimize it.

Sincerely,

Muzzamal Hussain et al.

Round 2

Reviewer 1 Report

Comments and Suggestions for Authors

In my opinion, the authors have appropriately improved the manuscript by taking into account my comments.

Reviewer 3 Report

Comments and Suggestions for Authors

I checked the previously published work by the authors, they mentioned that "The purity of arabinoxylans extracted from MB and NMB was 57.04 ± 0.7% and 60.1 ± 0.8%, respectively". Hence, the authors should add the purity of the FAX in materials and methods to avoid misunderstanding. 

The other comments have been addressed.